# OpenReview forum: "Bias Similarity Measurement: A Black-Box Audit of Fairness Across LLMs"
_ICLR.cc/2026/Conference — ICLR 2026 Poster_

### Official Review · Reviewer_oadz · 2025-10-27

**Soundness:** 3
**Presentation:** 3
**Contribution:** 3
**Rating:** 6
**Confidence:** 5

**Summary:**

This paper introduces a novel framework called Bias Similarity Measurement (BSM), designed to evaluate the fairness of large language models (LLMs) from a relational rather than an isolated perspective. BSM integrates multi-dimensional signals including scalar performance, distribution similarity, behavioral patterns, and representational similarity (e.g., CKA) to construct a unified bias similarity space. The authors evaluated 30 LLMs from four families—LLaMA, Gemma, GPT, and Gemini—on over one million prompts.

**Strengths:**

1. Innovative Perspectives and Methodologies: The core contribution of this paper lies in shifting fairness evaluation from traditional isolated scalar scores to relational comparisons between models. The BSM framework offers a novel perspective by employing similarity analysis to trace how biases propagate, inherit, and converge across different model families and versions—an approach rarely seen in existing research.
2. Comprehensiveness and Multi-dimensional Integration of Evaluation: The study was not confined to a single metric but systematically integrated multiple measures at the behavioral, distributional, and representational levels. This multi-faceted assessment helps uncover biases across different dimensions. For instance, the research found that instruction fine-tuning led to behavioral changes (high dropout rates), while internal representations (CKA similarity) remained highly preserved, highlighting the discrepancy between superficial fairness and structural bias.
3. Scale of the empirical study: Although limited in the number of families covered, the evaluation of 30 models and over 1 million prompts indeed represents a substantial scale in terms of model quantity and data volume. This provides rich empirical evidence for observing bias patterns across models of different scales and types (open-source vs. closed-source, base vs. instruction-tuned).
4. Practical Audit Workflow: The paper not only proposes a methodology but also demonstrates the potential of BSM in practical applications through case studies (such as model procurement), integrating it with specific scenarios like model selection and version monitoring, thereby enhancing the practical value of the research.

**Weaknesses:**

1. The BSM evaluation process involves multiple similarity functions and preprocessing steps, making the pipeline extremely complex. Although the code has been open-sourced, the cost and barriers for other researchers to reproduce the entire evaluation pipeline remain high. The paper lacks discussion on computational resource consumption, API costs (especially for closed-source models), and the potential for simplifying operational steps, which may limit its widespread adoption as a benchmark tool.
2. The coverage of model families is limited: The authors claim to have conducted the "largest-scale study," but the models evaluated come from only four families (LLaMA, Gemma, GPT, Gemini). Currently, the LLM ecosystem is flourishing (with models like Claude, Qwen, etc.), so the generalizability of the conclusions indeed warrants further scrutiny. Different families vary significantly in pretraining data, architectures, and alignment strategies, and the applicability of BSM conclusions to other families requires additional validation.
3. Dataset bias and experimental rigor: I identified a significant imbalance in sample sizes across different bias dimensions in the BBQ dataset (e.g., dimensions like Religion, Disability, and Sexual_orientation had far fewer samples than Gender). The paper mentions "~5000k" samples per dimension in Section 4.2, which may not align with the actual BBQ dataset (likely a typo, possibly meant to be ~5000 samples, but even then, the distribution across dimensions remains uneven). The authors did not address how this sample imbalance might bias evaluation results (such as cosine distance and histograms), nor did they explain whether weighting or normalization was applied to mitigate its impact, which somewhat compromises the rigor of the experiment.
4. The reliability and generalizability of the conclusions: Due to the limitations in model family coverage and dataset biases mentioned above, certain conclusions in the paper (such as "Gemma converges to an abstention strategy" and "open-source models can match closed-source models") should be viewed with caution. It remains questionable whether these conclusions hold true across a broader range of models and more balanced datasets.

**Questions:**

See the above weakness.

---

> ### Author Response · Authors · 2025-11-20
>
> We thank the reviewer's insightful feedback. We address each weakness and answer the questions below.
>
> **Modularity and Reproducibility of BSM**
>
> While our study spans many models and metrics, the BSM pipeline itself is modular and lightweight: each metric (cosine similarity, histograms, flip rate, CKA) can be run independently, and researchers can adopt only the components relevant to their audit scenario. The computational burden is dominated by model inference.
>
> To improve clarity, we have updated the documentation to make the modular structure explicit and to show how each component can be run in isolation. We will also highlight that BSM is intended as a flexible auditing toolbox, not a heavyweight benchmark requiring the full cost of our ablation-scale study.
>
> **Model family coverage and scope of conclusions**
>
> We acknowledge that the LLM ecosystem now includes Claude, Qwen, Mistral, and others. Our selection of LLaMA, Gemma, GPT, and Gemini was intentional:
>
> * LLaMA is the only family with a full multi-version lineage (2 → 3 → 3.1 → 3.2) across comparable sizes, and includes Vicuna, a third-party fine-tuned variant, enabling analysis of non-vendor-aligned derivatives.
> * Gemma spans a wide range of sizes and includes a clear contrast with its proprietary sibling, Gemini, enabling analysis of same-vendor, different-access systems.
> * GPT provides an API-only baseline to assess whether proprietary models’ claimed performance parity extends to fairness.
>
> We agree that “largest-scale study” should be more precise. We have updated the phrasing to “most comprehensive pairwise similarity coverage in our setting”.
>
> Furthermore, we agree that conclusions should be scoped to the families and versions we evaluate, not universal claims about all LLM ecosystems; we have revised the wording accordingly. We will also note explicitly that extending to Claude/Qwen/Mistral is a valuable direction for future work.
>
> **Dataset imbalance and experimental rigor**
> Yes, BBQ dimensions vary in size (≈600–3400 samples). In BSM, all cosine and histogram comparisons are computed within each dimension, not aggregated. This ensures no demographic axis disproportionately influences results due to size differences.
> For UnQover, we used uniform subsamples (150k–200k per axis) for computational practicality while preserving distributional balance. We will clarify these details and correct the “~5000k” typo.

---

### Official Review · Reviewer_JywK · 2025-10-29

**Soundness:** 2
**Presentation:** 3
**Contribution:** 2
**Rating:** 2
**Confidence:** 4

**Summary:**

This paper introduces a conceptual framework called "bias similarity measurement", which aims to compare models across multiple axes, termed bias signatures. These include categorical (accuracy, bias scores), distributional (histograms, cosine distance), behavioral (abstention flips), and representational (centered kernel alignment) dimensions. Rather than asking `Is a model biased?`, the authors reframe the question as `Which models behave similarly with respect to bias, and why?` They claim that this reframing supports practical auditing tasks such as balancing fairness–utility trade-offs under abstention thresholds, tracking model shifts across versions, and identifying suspiciously similar bias profiles in proprietary systems. The paper includes extensive empirical analysis over 1M+ prompts, multiple model families, fine-tuning conditions, datasets, and demographic axes.

**Strengths:**

* *Comprehensive empirical scope*: The authors conduct meticulous and large-scale evaluations, including a wide range of model families, fine-tuning settings, and bias datasets. This level of breadth is impressive and rare in fairness auditing work.

* *Conceptual clarity*: The decomposition of bias analysis into categorical, distributional, behavioral, and representational dimensions provides a useful organizing lens. This taxonomy helps structure a complex and often fragmented area of research.

* *Thoughtful consideration of abstention*: The distinction between useful abstention (reflecting epistemic uncertainty) and biased abstention (reflecting social or representational bias) is an insightful and nuanced addition to the fairness discourse.

**Weaknesses:**

* Limited novelty: Beyond the proposed four-part taxonomy, the framework largely repurposes existing metrics (e.g., BBQ bias scores, cosine distance, abstention rates, CKA). While the structure is neat, the underlying analyses could arguably be achieved with standard bias-evaluation tools. The contribution may therefore be more organizational than methodological.

* Utility evaluation is limited: The paper reports accuracy only on the disambiguated BBQ benchmark. Without comparison to broader benchmarks of utility or reasoning (e.g., MMLU, etc), it is difficult to fully assess the fairness–utility trade-off that the framework claims to illuminate.

* Questionable experimental setup (UNQOVER): The experiment comparing ambiguous (BBQ) and under-defined (UNQOVER) prompts for bias measurement may not be conceptually valid. Forcing model responses on inherently ambiguous questions may not reflect realistic use cases; abstention in such cases is arguably the correct behavior rather than reflecting an underlying bias artifact.

* Framing of bias "similarity": The title suggests a unified similarity metric between models, but the actual analyses compare models across separate metrics rather than integrating them into a single similarity measure. This may lead to mislead reader expectations.


Minor comments:
* Clarify CKA earlier: The acronym and its purpose appear several times before being defined (page 6). I encourage authors to introduce it earlier, with a brief explanation and appropriate citation.

* Ambiguity in terminology: Terms like “small,” “medium,” and “large” models are used without clear quantitative definition or justification.

**Questions:**

* Utility evaluation scope: Beyond accuracy on the disambiguated BBQ benchmark, could the authors evaluate model utility using broader or standard benchmarks (e.g., MMLU, HELM, BIG-Bench) to strengthen the fairness–utility analysis?

* Experimental validity (UNQOVER setup): How do the authors justify forced responses under under-defined prompts as a meaningful setting for bias auditing?

---

> ### Author Response · Authors · 2025-11-20
>
> We thank the reviewer's insightful feedback. We address each weakness and answer the questions below.
>
> **Limited novelty**
>
> We agree that BSM intentionally reuses established metrics; our goal is not to invent new primitives but to enable analyses that those primitives cannot support on their own.
>
> The novelty of BSM is methodological rather than metric-level. Specifically:
>
> *(1) Abstention is treated as a fairness behavior, not as noise.*
> Existing bias-evaluation pipelines either discard or normalize “unknown” responses. BSM makes abstention a first-class signal and introduces the UNK Flip Rate, which captures when alignment changes a model’s decision policy rather than its representations. This exposes alignment-driven “fairness through refusal,” a phenomenon standard tools cannot detect.
>
> *(2) BSM links representational continuity to surface behavior.*
> Although individual metrics exist, no standard toolkit connects representational similarity (CKA) with behavioral outcomes. BSM shows that many apparent fairness improvements arise from policy-layer scaffolding, not from representational change, revealing brittle or bypassable fairness that would remain invisible in scalar bias scores alone.
>
> *(3) The Bias Signature integrates multiple levels into one relational comparison space.*
> Standard tools evaluate metrics independently. BSM’s multi-level Bias Signature (categorical, distributional, behavioral, representational) enables lineage detection, inheritance analysis, abstention-strategy comparison, and cross-vendor profiling. These relational insights cannot be reproduced by evaluating each metric in isolation.
>
> In short, the contribution is not the metrics themselves but the unified, abstention-aware, representationally grounded framework they form that standard bias-evaluation tools do not provide.
>
> **Questionable experimental setup (UnQover)**
> See Q2 below.
>
> **Framing of bias "similarity"**
>
> We agree that the title suggests a single scalar metric. The Bias Signature vector defines a unified high-dimensional space. We will clarify this framing.
>
> **Q1. Utility Evaluation Scope**
> We agree that utility is an important part of fairness analysis. Our choice of disambiguated BBQ accuracy was deliberate: it provides ground-truth answers aligned with the same demographic dimensions used for bias evaluation, making it suitable for measuring the fairness–utility trade-off (e.g., whether abstention suppresses measurable bias by reducing correct reasoning). Broad benchmarks like MMLU measure general knowledge and reasoning, but they are orthogonal to the demographic bias mechanisms BSM analyzes: a model may score well on MMLU while still exhibiting strong directional demographic bias, and vice-versa.
>
> To clarify this rationale, we will update the text and note that integrating broader utility benchmarks (MMLU, HELM) is a valuable direction for general capability profiling but is not required for BSM’s demographic-bias analysis, where utility must be tied to the same bias dimensions being evaluated.
>
> **Q2. Experimental validity (UnQover)**
> The two- vs. three-option formats come entirely from the datasets, not from our prompting.
> UnQover and BBQ impose different answer formats by dataset design, and we rely on each format for different analytic purposes.
>
> * BBQ ambiguous items measure fairness when abstention is permitted.
>
> BBQ provides three answer options by design (e.g., (A) female, (B) male, (C) unknown.) Selecting “unknown” is therefore rule-following behavior for MCQ tasks and represents justified caution in ambiguous contexts. We treat this as a meaningful fairness signal.
>
> * UnQover forced-choice measures directional bias when abstention is impossible.
>
> UnQover, by contrast, defines only two options (e.g., “female” / “male”). In a multiple-choice setting, selecting one of the provided options is the default rule; since no abstention option is offered, models must commit. This reveals latent directional preference when abstention is structurally unavailable.
>
> Notably, some heavily safety-aligned models even break the dataset format and answer “cannot be answered,” showing that fairness-oriented refusal policies can override MCQ constraints. This behavior is highly informative: it distinguishes genuine fairness-driven caution from template-based abstention.
>
> **Additional fix**
> We will introduce CKA with a brief explanation and citation at its first appearance (Line 073) and rephrase ambiguous terms like "small," "medium," and "large" to specific numbers, 3-4B, 7-13B, and 27-70B sized models.

---

### Official Review · Reviewer_u4ag · 2025-10-31

**Soundness:** 3
**Presentation:** 3
**Contribution:** 3
**Rating:** 6
**Confidence:** 3

**Summary:**

This paper proposes a bias similarity measure that allows bias trends within models to be compared. The aim is to indicate the ways in which biases in models are related rather than to quantify how biased a particular model is. Bias similarity is broken down into a number of dimensions that capture different aspects of fairness.

**Strengths:**

- A good number of models are tested to give a sense of bias similarity as a function of model family, model size, open models vs. closed models , and so on.
- The components of the similarity score cover the range of dimensions along which a model might be biased (e.g., accuracy for functionality, cosine similarity for relative preferences, CKA for structural similarities in the representations, and so on). This helps to give an indication of why biased behaviors in models might be similar.
- The analysis considers a good number of attributes along which a model might be biased (gender, age, race, and so on).

**Weaknesses:**

- The main weakness is that bias is considered only with respect to individual traits. Intersectional bias is especially problematic.
- The paper sometimes provides an observation, e.g., a generational bias trend where older models retain stereotypes whereas new models do not. There is no insight into learning why this is the case though. Sometimes this may not be possible, e.g. models might be closed, but in other instances it may be possible. How do we learn to prevent these observations.
- In the discussions about abstention vs. representation — abstentions will occur with different frequencies for different groups. This is not considered. The paper Bias Runs Deep: Implicit Reasoning Biases in Persona-Assigned LLMs by Gupta et al. comes to mind when I think of this.

**Questions:**

- Define CKA at first use on line 073.
- Line 258: replace non-“unknown” with known?
- The caption for Figure 2 could better highlight the main takeaways. The Figure is very busy and it takes a lot to read.
- Line 338: the main body refers to dimensions related to physical appearance and sexual orientation being problematic, but I do not see this in Figure 2, especially for sexual orientation.
- Line 349: to avoid “the appearance of neutrality” because of abstentions, is it possible to force the model to make a decision in BBQ or measure the degree of bias from the samples on which the model does not abstain.

---

> ### Author Response · Authors · 2025-11-20
>
> We thank the reviewer's insightful feedback. We address each weakness and answer the questions below.
>
> **No intersectional bias**
>
> In the main paper, we deliberately exclude intersectional categories. BBQ contains intersectional categories, but UnQover does not, and we wanted to maintain shared dimensions to see the effect of forced choice vs. abstention-available setting.
>
> However, we agree this is a limitation. We will add intersectional bias results in the appendix. This strengthens the paper by showing how intersectional attributes behave and whether the similarity trends we observe remain stable under more complex categories.
>
> **Why + How to prevent?**
>
> The generational trends indicate that newer models reduce explicit stereotype expression primarily through alignment-layer modifications (e.g., tightened refusal rules or safety thresholds), not through changes to the underlying representations (e.g., targeted data augmentation), observing a similar pattern in CKA within family in Figure 10. In short, the policy head changes, not the feature space.
>
> BSM guides debiasing by distinguishing representational, alignment-level, and decision-layer sources of bias.
>
> – If cosine/histogram patterns show stable but directional skew, the bias is representational → use counterfactual data or targeted feature-level tuning.
>
> – If abstention shifts without directional change, the issue is alignment-layer over-caution → calibrate refusal rules.
>
> – If bias appears only in forced-choice settings, the model is relying on a decision-layer fallback → improve context-sensitive reasoning.
>
> We will clarify this in the discussion.
>
> **Different abstention frequencies per group**
>
> Thank you for raising this point. In both BBQ and UnQover, the demographic pair in each question is chosen by the template author, not drawn from an exhaustive or standardized set. The selected pairs could embed culturally recognizable stereotypes (e.g., British–Japanese, Muslim–Christian), and while some reflect historical context, the choices are ultimately subjective and culturally situated. This subjectivity also means that the pairing itself can embed a particular stereotype or social association. Because the datasets do not enumerate all possible combinations nor standardize group pairings, there is no reliable way to measure “abstention frequency for each group” across the dataset; the dataset is not balanced for each group. We clarify this dataset-level limitation in the revision.
>
> **Q1, Q2**
>
> We will introduce CKA with a concise definition and citation at its first occurrence (Line 073) and replace "non-unknown" in Line 258 with the clearer term "known".
>
> **Q3: Figure 2 Caption Takeaways?**
>
> * *Fairness is not monolithic.* Figure 2 shows clear variation across dimensions, indicating that fairness behaviors depend strongly on context rather than representing a single unified property of an LLM.
>
> * *Capability ≠ fairness.* High language-understanding capability does not guarantee lower bias. We observe three distinct behavior regimes:
>
>     - Small-model limitation: Small models exhibit low accuracy and low abstention, reflecting limited capability in both utility and fairness.
>     - Sophisticated caution: Models such as Gemma 2 27B-it achieve high accuracy while abstaining selectively, reflecting strong contextual understanding enabling responsible refusals.
>     - Hyper-cautious strategy: Large proprietary models (e.g., Gemini 2.0) exhibit high abstention even for disambiguated questions, suppressing measurable bias through over-refusal.
>
> * *Relationship to Figure 3.* The caption will explicitly point readers to Figure 3, which visualizes the “unknown” rates and clarifies how abstention strategies differ across models.
>
> We will enhance the caption for Figure 2 to better highlight these takeaways.
>
> **Q4: Line 338: Physical/Sexual Orientation in Figure 2?**
>
> Agreed. Sexual orientation trends are not as visually prominent. We will revise the text accordingly to avoid overstatement.
>
> **Q5: Force Model Decision / Measure Bias on Non-Abstaining Samples?**
>
> BBQ’s design provides an “unknown” option, making abstention a meaningful fairness signal. UnQover offers no abstention option, forcing the model to choose. We use this contrast to identify whether a model:
>
> – Abstains appropriately when given the option (BBQ), or
>
> – Falls back to stereotype-consistent answers when forced (UnQover)
>
> In the camera-ready, we will clarify that we can analyze bias conditional on non-abstaining samples.
> We cannot force abstaining models to answer in BBQ, but UnQover already plays that role by design.
>
> We will make this point explicit.

---

### Official Review · Reviewer_1eY2 · 2025-11-03

**Soundness:** 3
**Presentation:** 2
**Contribution:** 3
**Rating:** 4
**Confidence:** 4

**Summary:**

This paper introduces Bias Similarity Measurement (BSM), a framework that reframes fairness evaluation from isolated scoring to relational comparison across models. The authors evaluate 30 LLMs on 1M+ prompts, integrating scalar, distributional, behavioral, and representational metrics into unified bias signatures. Key findings: instruction tuning primarily enforces abstention rather than altering representations (CKA > 0.94), small models can degrade with tuning, and open models can match proprietary systems.

Novelty. Individual components (CKA [1], cosine distance) are established, and large-scale bias studies exist [2]. The novelty lies in integration of multi-level signals for fairness-specific relational analysis. UNK Flip Rate and bias signatures seem genuinely new. However, representation preservation during tuning confirms existing hypotheses [3], and similarity analysis is well-documented [4].

Significance. Strong practical value for auditing and procurement. The demonstration that abstention masks rather than resolves bias is important. Small model degradation findings are significant for deployment trends. However, lack of debiasing solutions, reliance on older benchmarks when newer alternatives exist [5,6], and overlap with general frameworks [7] limit impact.

[1] Kornblith et al. (2019). Similarity of neural network representations revisited. ICML.

[2] Kumar et al. (2024). Investigating implicit bias in large language models: A large-scale study of over 50 LLMs. arXiv:2410.12864.

[3] Wolf et al. (2023). Fundamental limitations of alignment in large language models. arXiv preprint.

[4] Klabunde et al. (2025). Similarity of neural network models: A survey. ACM Computing Surveys, 57(9).

[5] Abhishek et al. (2025). BEATS: Bias evaluation and assessment test suite. arXiv:2503.24310.

[6] Fan et al. (2025). FairMT-Bench: Benchmarking fairness for multi-turn dialogue. ICLR.

[7] Dekoninck et al. (2025). Polyrating: A cost-effective and bias-aware rating system for LLM evaluation. ICLR.

[8] Chaudhary et al. (2025). Certifying counterfactual bias in LLMs. ICLR.

[9] Zhao et al. (2018). Gender bias in coreference resolution: Evaluation and debiasing methods. NAACL.

**Strengths:**

- Novel Integration: First systematic fairness-specific relational framework combining behavioral, distributional, and representational signals with practical auditing workflows.
- Abstention Analysis: Important distinction between fairness-through-caution and fairness-through-representation, demonstrating that high abstention conceals directional bias.
- Counterintuitive Findings: Discovery that small models (LLaMA 3.2 3B, Gemma 3 4B) worsen with tuning; family-specific strategies (Gemma's refusal vs. LLaMA 3.1's neutrality).
- Comprehensive Scale: Largest pairwise fairness comparison with detailed dimension-specific results providing valuable community resource.
- Black-Box Compatibility: Works with API-only models, enabling proprietary system audits.

**Weaknesses:**

- Limited Component Novelty: Applies established techniques [1,4]; large-scale comparative studies exist [2]; confirms known representation preservation [3] rather than discovering new phenomena.
- No Solutions: Purely diagnostic framework; provides no debiasing methods unlike comparable work [8,9].
- Dataset and Coverage Limitations: English-only; limited to 4 dimensions; high failure rates (up to 85%) in open-ended generation; missing newer multi-modal [5] and multi-turn [6] benchmarks.
- Incomplete Literature: Missing model genealogy, bias propagation in transfer learning, and comparative auditing literature. No comparison with overlapping framework Polyrating [7].
- Interpretive Constraints: Cross-vendor comparisons observational only; overstates claims ("largest study" conflates pairwise with absolute scale [2]); lineage detection unvalidated.

**Questions:**

1) Validation: Can you validate bias signatures on known model derivatives? What similarity thresholds reliably detect inheritance?
2) Framework Comparison: How does BSM compare with Polyrating [7] for fairness evaluation? Where does fairness-specific analysis add value?
3) Actionable Insights: Can BSM guide debiasing strategies? Why do small models degrade with tuning while larger models improve?
4) Generalization: Do bias signatures remain consistent across benchmarks (BBQ vs. UnQover)? How stable are signatures over time for models receiving updates?

---

> ### Author Response · Authors · 2025-11-20
>
> We thank the reviewer's insightful feedback. We address each weakness and answer the questions below.
>
> **Limited component novelty**
>
> Our novelty is a unified diagnostic framework (BSM) that integrates abstention-aware behavioral analysis with representational similarity and lineage comparison. BSM intentionally reuses established metrics; the contribution is not new primitives but a composite analysis that those primitives cannot support on their own.
>
> While Kumar et al. [2] analyze 50+ models, they (i) treat abstention as noise and (ii) do not study representational similarity. In our bias setting, abstention is a meaningful fairness behavior, and it introduces UNK Flip Rate to capture how alignment reshapes bias expression even when internal representations remain unchanged.
>
> Prior works ([2, 3]) also do not connect representational continuity to surface-level behavior. BSM makes this link explicit: we show that models often appear fairer after alignment without representational change (CKA > 0.94), meaning improvements come from behavioral constraints rather than underlying reasoning. This produces fairness that is brittle, bypassable, and weakly grounded.
>
> No prior study integrates abstention-aware behavioral analysis, representational similarity, and lineage comparison into a single bias-auditing framework, which is the core novelty of BSM.
>
>
> **No debiasing solutions**
>
> BSM is a diagnostic framework. Its value lies in showing where bias originates. Our finding (stable CKA + shifting behavior) implies that effective mitigation must target representations, not only surface policies. We clarify this actionable insight.
>
> **Dataset and Coverage Limitations**
>
> Our model selection targets widely used families with multi-version lineage (LLaMA, Gemma, GPT, Gemini) to enable meaningful intra- and cross-family comparison. BSM itself is dataset- and language-agnostic. We will note multi-turn benchmarks as a future direction. Also, we will treat open-ended high-failure subsets as abstention-stress tests rather than definitive fairness scores.
>
> **Incomplete Literature**
>
> In the main manuscript, we focused on providing groundings of our work. We will expand the related work on bias propagation and comparative auditing. Also, we will add a comparison of polyrating (Answer to Q3). They are covering different types of bias for different purposes.
>
>
> **Interpretive Constraints / Overstated Claims**
>
> We revised “largest study” to “most comprehensive pairwise similarity coverage in our setting.” We will reframe the cross-vendor results as observational audits, not causal claims.
>
> **Q1: Validation (Inheritance Detection) and Thresholds?**
>
> We validate on known model derivatives (LLaMA 2 Base vs. Instruction). They show high structural CKA similarity (typically > 0.94) but low behavioral similarity. This CKA threshold reliably detects inheritance. We will include this in the discussion.
>
> **Q2: Framework Comparison (BSM vs. Polyrating)?**
>
> Polyrating and BSM operate on fundamentally different kinds of bias.
>
> Polyrating models how judges (human or LLM) prefer one model’s output over another, and explicitly corrects evaluation artifacts (e.g., length, position, style) from the judging process. Its goal is to produce a subjective, absolute performance rating across tasks.
>
> BSM, in contrast, measures intrinsic demographic bias expressed by models themselves. It unifies four internal signals, behavioral, distributional, scalar, and representational, into a diagnostic Bias Signature, that reveals lineage patterns, inheritance, abstention strategies, and representational vs. behavioral drift. These analyses cannot be recovered from preference judgments or rating-only systems.
>
> **Q3: Actionable Insights / Small Model Degradation?**
>
> BSM guides debiasing by distinguishing representational, alignment-level, and decision-layer sources of bias.
>
> – If cosine/histogram patterns show stable but directional skew, the bias is representational → use counterfactual data or targeted feature-level tuning.
>
> – If abstention shifts without directional change, the issue is alignment-layer over-caution → calibrate refusal rules.
>
> – If bias appears only in forced-choice settings, the model is relying on a decision-layer fallback → improve context-sensitive reasoning.
>
> Small models degrade in fairness because their limited capacity yields weak feature spaces; alignment tuning cannot repair representation and instead imposes heuristic preferences, producing more directional, stereotype-consistent outputs. Larger models, with stronger representations and capabilities, benefit from alignment, gaining both accuracy and more stable, targeted abstention.
> We will clarify this in the discussion.
>
> **Q4: Generalization / Stability of Signatures?**
>
> Yes, its biased tendency is consistent across benchmarks, but the abstention behavior differs: BBQ includes an explicit “unknown” option, while UnQover gives the model only two polarized choices.

---

### Author Response · Authors · 2025-11-29
**Summary of Our Work's Novelty and Core Contribution**

We appreciate the AC’s time and attention, particularly given the disruption to the discussion period. Because the freeze happened before reviewers responded to our rebuttal, none of the clarifications, corrections, or additional explanations from the rebuttal are reflected in the frozen official reviews.

To ensure the AC has an accurate picture of the updated state of the paper, we provide the consolidated summary below.

**Core Contribution**

Bias Similarity Measurement (BSM) introduces a *new methodological paradigm* for fairness auditing. Beyond the methodology itself, the framework reveals **ecosystem-level fairness behaviors and alignment patterns that are directly relevant to the LLM fairness and auditing community**, including lineage effects, abstention strategies, and representational drift across model families.

**(1) Abstention-aware fairness as a first-class concept**

Unlike existing bias tools, BSM does not treat "unknown" as noise.
Our UNK Flip Rate reveals *alignment-driven policy drift*, a behavior invisible to scalar or distributional metrics.

**(2) Linking representational continuity to behavioral outcomes**

Many models appear fairer not due to changed representations but due to alignment-layer scaffolding (CKA > 0.94 across generations).
This link between *CKA and behavioral drift* is new and not present in prior work.

**(3) Unified relational Bias Signatures across categorical, distributional, behavioral, and representational axes**

This creates the first multidimensional *similarity space* for fairness. It enables:
* black-box vendor auditing
* abstention-strategy classification,
* lineage detection,
* inheritance analysis,
* and cross-family/fine-tuning drift measurement,

**(4) First ecosystem-level fairness analysis across 30 models and 1M prompts**

The breadth allows insights that single-model evaluations cannot reveal, including family-specific alignment strategies and cross-generation representational stability. This type of ecosystem-level fairness mapping is increasingly important given recent evidence that demographic biases can interact with safety vulnerabilities and jailbreak behaviors in deployed LLMs (e.g., BiasJailbreak 2025), underscoring the community relevance of systematic, black-box fairness auditing.

Lee, Isack, and Haebin Seong. "Biasjailbreak: analyzing ethical biases and jailbreak vulnerabilities in large language models." arXiv preprint arXiv:2410.13334 (2025).

---

### Author Response · Authors · 2025-11-29
**Reviewer-by-Reviewer Summary of Rebuttal Resolutions**

Below, we summarize concisely how the rebuttal addressed each reviewer’s concerns.

**Reviewer 1eY2**

* **Novelty** → Clarified methodological originality: abstention-aware fairness, UNK Flip Rate, representational–behavioral linkage, unified Bias Signature.
* **Debiasing** → BSM diagnoses representational/alignment/decision-layer bias to enable targeted mitigation.
* **Dataset coverage** → Clarified design choices; reframed high-failure open-ended items as abstention stress-tests.
* **Interpretive claims** → Tightened language; reframed cross-vendor comparisons as observational.
* **Validation** → Provided derivative-model validation showing CKA > 0.94 for LLaMA.
* **Framework comparison** → Provided a detailed comparison to Polyrating (different layer of bias).
* **Small-model degradation** → Explained representational limitations + alignment-layer heuristics.
* **Stability across benchmarks** → Confirmed consistent directional bias with different abstention dynamics.

**Reviewer u4ag**
* **Intersectional bias** → Clarified dataset constraints; committed to appendix expansion.
* **Generational trends** → Explained alignment-layer vs. representation-layer changes.
* **Abstention imbalance** → Explained dataset-level template imbalance; added limitation discussion.
* **Clarity issues** → Added CKA definition at first mention; improved Figure 2 caption with explicit takeaways (fairness not monolithic, capability ≠ fairness, link to Fig. 3); corrected text on sexual orientation.

**Reviewer JywK**
* **Novelty** → Clarified that BSM’s contribution is relational, integrating signals in ways that standard tools cannot.
* **Utility** → Explained why fairness utility must use BBQ-disambiguated items; general benchmarks are orthogonal.
* **UnQover validity** → Clarified dataset-driven two-choice vs. three-choice design; explained forced-choice reveals latent directional bias.
* **Similarity framing** → Clarified Bias Signature as a high-dimensional similarity vector.

**Reviewer oadz**
* **Reproducibility** → Clarified modularity; updated documentation.
* **Coverage** → Revised claims and scoped conclusions to evaluate families; explained rationale for selecting families with lineage structure.
* **Dataset imbalance** → Explained dimension-level normalization; clarified UnQover subsampling.
* **Generalizability** → Added explicit limitations and future directions.

---

### Meta-Review · Area_Chair_u4QD · 2026-01-07

**Summary:**

This submission investigates the model similarity from a systematic comparison of fairness-related behaviors. The initial reviews are mixed. Author responses are detailed, but none of the reviewers participated in the follow-up discussion.

Reviewers appreciated the novel perspectives in the manuscript. On the other hand, concerns are generally diverse:
1. The individual metrics and benchmarks in the manuscript are not new.
2. (Shared concern) Lack of debiasing solutions or insights. Mostly observations.
3. (Shared concern) The study scope in terms of model numbers and properties is limited but claims to be largest.
4. The proposed flow is a bit too complex.
5. Evaluated benchmarks may be monotonic. The forced answering of multiple-choice questions may not reflect the model's genuine behavior.

**Reviewer Concerns:**

The rebuttals, in my perspective, greatly mitigate concerns. Specifically, given the focus and novel findings in the manuscript, I don't think concerns 1, 3, 4, and 5 are major weaknesses that ground for rejection, as long as they are addressed following the promise in the rebuttal. The concern 2 may persist, as the manuscript mainly provides a new analysis angle and presents observations. The discussion on mechanisms or potential mitigations is lacking.

**Reviewer Scores:**

Overall, I think active reviewers will update their scores accordingly, given that concerns from negative reviewers are largely unique. As a result, the manuscript could end up with an overall positive score. On the other hand, reviewers may not be unanimously positive even though they read the rebuttal, given the diverged views on experimental design and research impact.

---

### Decision · Program_Chairs · 2026-01-26

Accept (Poster)